# Mapping combinatorial drug effects to DNA damage response kinase inhibitors

Hanrui Zhang [1], Julian Kreis[2], Sven-Eric Schelhorn[2], Heike Dahmen[2], Thomas Grombacher[2], Michael Zühlsdorf[2], Frank T. Zenke [2] & Yuanfang Guan [1,3] ✉

One fundamental principle that underlies various cancer treatments, such as traditional chemotherapy and radiotherapy, involves the induction of catastrophic DNA damage, leading to the apoptosis of cancer cells. In our study, we conduct a comprehensive dose-response combination screening focused on inhibitors that target key kinases involved in the DNA damage response (DDR): ATR, ATM, and DNA-PK. This screening involves 87 anti-cancer agents, including six DDR inhibitors, and encompasses 62 different cell lines spanning 12 types of tumors, resulting in a total of 17,912 combination treatment experiments. Within these combinations, we analyze the most effective and synergistic drug pairs across all tested cell lines, considering the variations among cancers originating from different tissues. Our analysis reveals inhibitors of five DDR-related pathways (DNA topoisomerase, PLK1 kinase, p53-inducible ribonucleotide reductase, PARP, and cell cycle checkpoint proteins) that exhibit strong combinatorial efficacy and synergy when used alongside ATM/ATR/DNA-PK inhibitors.

Cancers are aggressive, invasive diseases characterized by uncontrolled growth. Many cancers exhibit genome instability resulting from tumor-specific DNA repair defects and increased replication stress, making them more susceptible than normal tissues to DNA damage, such as single and double strand breaks (SSBs and DSBs, respectively)[1,2]. Taking advantage of this vulnerability, DNA-damaging treatments such as ionizing radiation and platinum-based antineoplastic have long been used as anti-cancer treatments[3,4]. More recently, a suite of therapeutic agents targeting DNA damage response (DDR) pathways has been developed that specifically exploits this susceptibility, promising reduced side effects compared to non-targeted treatments[5–8]. In this context, it is hypothesized that the simultaneous deactivation of multiple DDR pathways could lead to improved treatment efficacy by addressing both acquired treatment resistance and buffering by parallel DDR pathways[2].

A set of 450 proteins involved in different pathways of the DNA damage response has recently been mapped[9]. While it is commonly assumed that specific pathways exist that address different types of

DNA damage, e.g., for SSBs, DSBs, or mismatch repair, loss of function of a DDR pathway can be compensated by parallel repair pathways[2,10]. The simultaneous inhibition of multiple complementary DDR pathways by somatic mutation in the tumor and/or one or more targeted treatments, such as the synthetic lethality between PARP1 inhibition and BRCA1 loss of function[11,12], was therefore identified as a promising therapeutic strategy in clinical cancer treatments. This strategy also inspired the development of combination treatments of multiple DDR inhibitors to overcome resistance to single drugs, achieve synergistic effect, and expand DDR drugs' usage to other indications beyond BRCA-deficient cancers[8,13].

Three canonical DNA damage-sensing kinases that are central to the human DDR are ataxia telangiectasia mutated (ATM), ataxia telangiectasia and Rad3-related (ATR), and DNA-dependent protein kinase (DNA-PK), which is also referred to as protein kinase, DNA-activated, catalytic subunit (PRKDC)[14,15]. So far, studies that comprehensively map the synergistic effects between small molecule inhibitors of these key DDR kinases and other anti-cancer drugs are lacking

[1]Department of Computational Medicine and Bioinformatics, Michigan Medicine, University of Michigan, Ann Arbor, MI, USA. [2]Merck Healthcare KGaA, Darmstadt, Germany. [3]Department of Internal Medicine, Michigan Medicine, University of Michigan, Ann Arbor, MI, USA. ✉e-mail: gyuanfan@umich.edu

in both coverages across tumor types and the number of combination therapy partners. In this study, we generated cancer cell line drug combination screens of six kinase inhibitors, including two ATM inhibitors (M3541 and M4076[16]), three ATR inhibitors (berzosertib[17,18], gartisertib[19], M1774[20]), and one DNA-PK inhibitor (peposertib[21,22]) against 87 anti-cancer drugs of a wide range of mode-of-actions on 22 - 62 cancer cell lines across 12 tissues (or tumor types), forming a total of 17,912 combination treatment experiments.

In order to characterize tissue-specific patterns of DDR inhibitor combination treatments, we carried out full-genome and transcriptomic profiling of all 62 cell lines and statistically associated dose responses with genomic and transcriptomic readouts. This screen represents a large DDR inhibitor combination study and allowed us to identify a small set of inhibitors to proteins involved in five pathways that displayed strong co-therapeutic efficacy and synergy with ATM/ATR/DNA-PK inhibition globally: the DNA topoisomerase pathway, the serine/threonine-protein kinase PLK1 pathway, the p53-inducible ribonucleotide reductase pathway, the PARP pathway, and the cell cycle checkpoint proteins.

## Results

### The experimental dose-response screen of three DDR inhibitors across a wide range of anti-cancer combination treatments

The goal of this study was to comprehensively analyze the synergistic relationship between the inhibitors of canonical DDR kinases (ATM, ATR, and DNA-PK) and a panel of anti-cancer drugs. In total, we combined six kinase inhibitors, including two ATM inhibitors (M3541 and M4076), three ATR inhibitors (berzosertib, gartisertib, M1774), and one DNA-PK inhibitor (peposertib) with 87 anti-cancer drugs, on 62 cancer cell lines covering 12 tissues or tumor types (Fig. 1a, Supplementary Data 1 and "Data availability"). For each of the cell lines, we carried out RNA- and whole-genome DNA sequencing, and derived

genome-wide readouts covering gene expression, copy-number profiling, and loss-of-function mutation both for single genes as well as biological pathways.

In vitro combination treatment responses were quantified on the level of both efficacy and synergy. The efficacy of treatment was estimated by the area above the parametric dose-response curve divided by the sum of the areas above and below this curve, a quantity that we denote as relative *AoC score*. The synergy between two combination partners within treatment was measured by the *Bliss score*, which reflects the additional effect of two drugs over the expected response if the two drugs were to act independently (see "Methods" section for detailed discussions of the dose-response experimental setup, cell line sequencing, and computation of response measures).

In total, we generated 17,912 combination treatment experiments and 7081 monotherapy experiments, with reproducibility of Pearson's correlation = 0.8380 ($p < 1e−22$) in AoC score for monotherapy and 0.7611 ($p < 1e−22$) in Bliss score for combination treatment, which is comparable with previously reported combination treatment screening datasets including DREAM[23], ALMANAC[23,24], and O'Neil[25]. While various DDR inhibitor combinations were used in our screens, we report results on the level of *mode-of-action combination* (e.g., ATMi-PARPi) for conciseness and generalizability. However, all analyses were conducted using and are supported by *individual drug combinations* (such as M3541-olaparib).

### Mapping the global interaction relationships between DDR inhibitors and combination treatment partners

In anti-cancer treatment, ideal drug combinations are not only safe and effective, but also complement each other in a synergistic manner[8]. Due to the complex relationships between DDR pathways[2], finding optimal drug combinations that show broad efficacy across multiple tumor types and genomic contexts of tumors is particularly

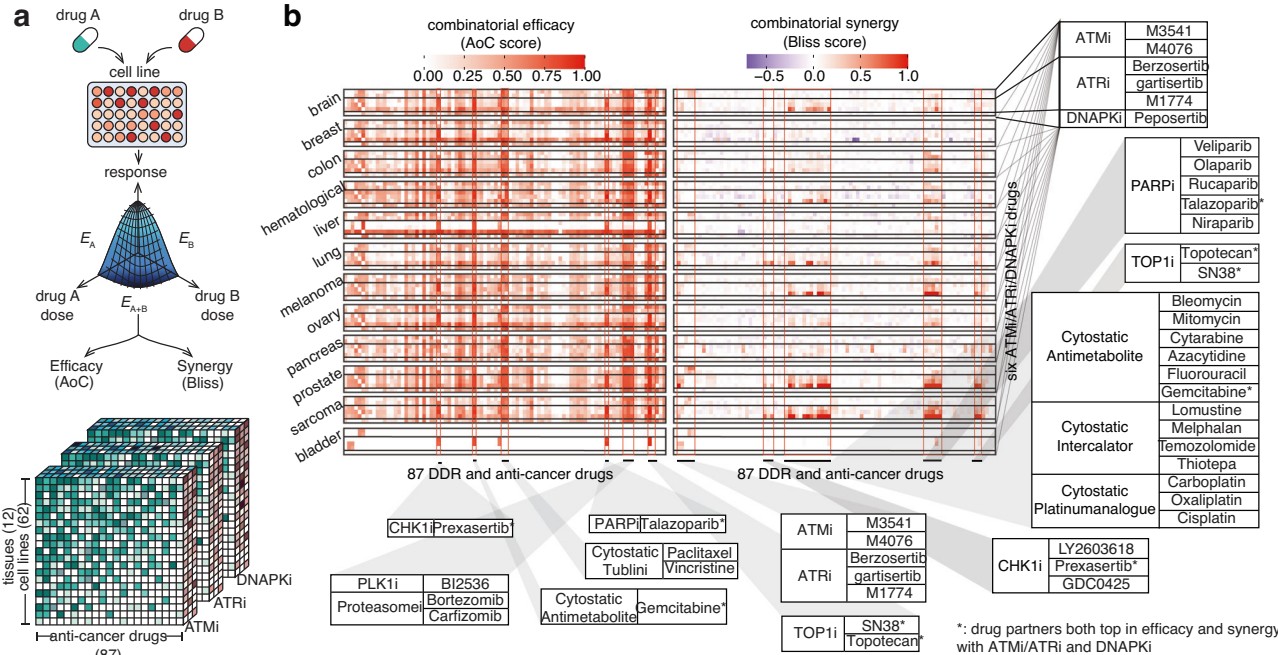

**Fig. 1 | Overview of combination treatment synergy screening experiments. a** Dose-response curves were used to calculate drug pairs' efficacy and synergy scores. Inhibitors to DDR kinases ATM, ATR, and DNA-PK (ATMi/ATRi/DNAPKi) were tested against 62 cell lines across 12 tissues. **b** DDR inhibitor combination treatment screens show strong interactions between drugs targeting different DDR factors. The efficacy (left panel, by area over the curve (AoC) score) and synergistic (right panel, by Bliss score) responses of all combination treatments across the 12

tissue types tested in this study are shown. Six DDR inhibitors of interest of three mode-of-actions (ATMi, ATRi, and DNA-PKi, shown on y-axis) combined with 87 drugs (x-axis), form 546 different combinations, which are facetted by the 12 different cancer cell line tissues of origin. Some drugs (and their mode-of-actions) with significant synergistic effects, when combined with the six DDR inhibitors of interest, are marked and shown in pop-out tables. More detailed information on all drug/mode-of-action combinations is shown in Supplementary Figs. 3 and 4.

challenging. This large-scale screen, therefore, provides a unique opportunity to map the overall global efficacy and synergy relationships between DDR inhibitors and other anti-cancer agents.

To visualize these global relationships, we generated comprehensive heatmaps showing the efficacy and synergy responses of all 87 anti-cancer drugs screened in combination with six ATM/ATR/DNA-PK inhibitors across all 62 cell lines and 12 tissues (Fig. 1b and Supplementary Figs. 1–3). By visual and numerical analysis, we identified several drugs that result in high efficacy when combined with ATM, ATR, and DNA-PK inhibitors. In general, ATR inhibitors have stronger synergy and efficacy compared to other DDR inhibitors in all combinations tested. In terms of the combination partners, tubulin inhibitors achieved high efficacy but low synergy with DDR inhibitors, possibly due to the high cytotoxicity of tubulin inhibitors alone[26] that may result in a plateau effect in cell growth inhibition which could not be further increased by combination with DDR inhibitors. Combination treatments with PARP inhibitors, such as veliparib, talazoparib, rucaparib, olaparib, and niraparib, which, with the exception of veliparib, are approved as targeted drugs for BRCA-mutated cancer treatment[8,27], demonstrated the highest synergy with ATM and ATR inhibitors across multiple cancer types. The TOP1/2 (DNA topoisomerase 1/2) inhibitors SN-38 (the active metabolite of irinotecan), topotecan, etoposide, and doxorubicin, also display high efficacy and synergy with ATM/ATR/DNA-PK inhibitors (DNA-PK > ATR > ATM), as previously reported in preclinical studies[19,28,29]. Last, selected chemotherapeutics such as gemcitabine, an antimetabolite that inhibits DNA synthesis, also achieved high efficacy and synergy when combined with ATR and ATM inhibitors (Fig. 2a, b). While the synergistic relationship between ATRi and gemcitabine has been reported before[30], we note that similar relationships between gemcitabine and either DNA-PKi or ATMi have not been reported before, to our knowledge. Overall, the dataset shows a low Pearson's correlation of 0.2 ($p < 1e{-}22$) between efficacy and synergy, which, while well within the range of values observed in previous studies[31,32], highlights the need of analyzing both measures of response independently.

In addition to analyzing results on the level of individual drugs, we further characterized the most efficacious and synergistic combination treatments identified in our screen by their mode-of-actions. Hierarchical clustering based on responses in different cell lines shows treatments with the same mode-of-actions tend to cluster together (Supplementary Figs. 3–7). For example, for monotherapy, ATM inhibitors (M3541 and M4076), CHK1 inhibitors (GDC0425 and LY2603818), and BET inhibitors (IBET151, CPI0610, and GSK525762A) are located adjacent to each other (Supplementary Fig. 1). The same pattern, i.e., combinations with the same or similar mode-of-actions are more likely to cluster together, also appears in combination response in terms of efficacy (Supplementary Figs. 4 and 5) and synergy (Supplementary Figs. 6 and 7). When combined with ATM, ATR, and DNA-PK, several modes of action consistently showed high efficacy and synergy (Fig. 2c, also see Supplementary Data 2), in particular TOP1i[33], RRM2Bi (the small subunit of p53-inducible ribonucleotide reductase)[34,35], PLK1i (polo-like kinase 1)[36], and checkpoint inhibitors CHEK1i and CHEK2i, suggesting that targeting cell cycle checkpoint may confer a significant benefit in the combination setting as has recently been suggested for ATRi-CHK1i[37].

Drug mode-of-actions identified from synergy analyses alone partly overlapped with those for efficacy scores; inhibiting RRM1/2 and TOP pathways seems to be broadly effective in combination with ATR/ATM/DNA-PK inhibition. The inhibition of RRM 1/2 pathway is only synergistic in combination with ATR, but not ATM and DNA-PK inhibition, while inhibiting TOP pathway is synergistic with all ATR, ATM and DNA-PK inhibition. Lastly, PARP inhibitors appeared to be strongly and broadly synergistic in combination with ATRi/ATMi, but not DNA-PKi (Fig. 2d and Supplementary Data 3).

## Four monotherapy and two DDR inhibitor combinations show significant variability in response between different cancer types

For investigating whether general biological backgrounds, such as cancer or tissue types, influence treatment response, we carried out statistical comparisons of the efficacy and synergy responses between different cancer types covering the 87 monotherapy agents and 465 combination treatments screened in our study.

As the number of cell lines covering each of the 12 cancer types varies, we chose the non-parametric Kruskal-Wallis test to analyze the variance of treatment response of each treatment across all cancer types in this study. After multiple testing correction, only four out of the 87 monotherapy agents showed significant variance in efficacy across different cancer types ($p < 0.01$), including doxorubicin ($p = 2.8e{-}08$), M3541 ($p = 2.2e{-}06$); peposertib ($p = 1.3e{-}05$), and oxaliplatin ($p = 3.4e{-}05$)) (Supplementary Fig. 1). Analogously, only two combination treatments out of the 465 combinations we tested showed significant variation in response across different cancer types: peposertib-gamma-ionizing-radiation (a DNA-PKi-IR combination showing significant cross-cancer type variance in terms of both efficacy ($p = 3.38e{-}3$) and synergy ($p = 7.82e{-}5$)), and M4076-berzosertib (an ATMi-ATRi combination showing variance only in terms of synergy ($p = 2.39e{-}05$)) (Fig. 3a, b). As in the results on the raw efficacy and synergy values (see previous sections), also no correlation of cross-cancer variance significance values between efficacy and synergy scores was detected (Pearson's $r = -0.028$, $p = 0.54$) (Fig. 3c), indicating again that the two scores are measurements of different pharmaceutical properties.

For all monotherapy and combination therapies that showed significant differences in responses across cancer types, we carried out statistical post hoc analysis including Dunn's test, to identify individual cancer types with variable responses to individual drugs and drug combinations (Fig. 3d–f, Supplementary Fig. 2 and Supplementary Data 4 and 5). Of the four significantly variable mono-therapeutic agents, doxorubicin showed significantly higher efficacy in hematological cancers than other cancer types, while M3541 demonstrated lower efficacy in both pancreas and melanoma cancers than other cancer types (Supplementary Fig. 2b). For peposertib and oxaliplatin, the difference of efficacy was only significant between bladder and ovary/hematological cancers, as well as between sarcoma and hematological cancers (Supplementary Fig. 2b). For the combination treatments, the peposertib-gamma-ionizing-radiation combination displayed significantly higher efficacy in hematological cancers compared to bladder cancers (Fig. 3d, e). Last, in the case of M4076-berzosertib, shows a significantly lower synergy in hematological cancers compared to pancreas, prostate, melanoma, and sarcoma cancers were observed (Fig. 3f). Interestingly, no significant correlation between average treatment efficacy or synergy and the significance of variance in different cancer types (across monotherapies (Pearson's $r = -0.01$, $p = 0.92$ and Spearman's $r = -0.075$, $p = 0.478$) and combination therapies, as well as for both efficacy (Pearson's $r = 0.01$, $p = 0.835$ and Spearman's $r = 0.01$, $p = 0.8$) and synergy (Pearson's $r = -0.04$, $p = 0.37$ and Spearman's $r = -0.021$, $p = 0.64$)) could be identified, indicating that the cancer type specificity and overall average treatment response are independent pharmaceutical characteristics.

## Discussion

We present a comprehensive combination treatment screening dataset focusing on DDR inhibitors, which allows us to identify interactions between DDR inhibitors and a broad range of anti-cancer drugs and map the molecular dependencies of their relationships. DDR inhibitors are an increasingly important class of targeted therapies explored for the treatment of cancer, and the results will help inform and recommend effective treatments depending on available genomic

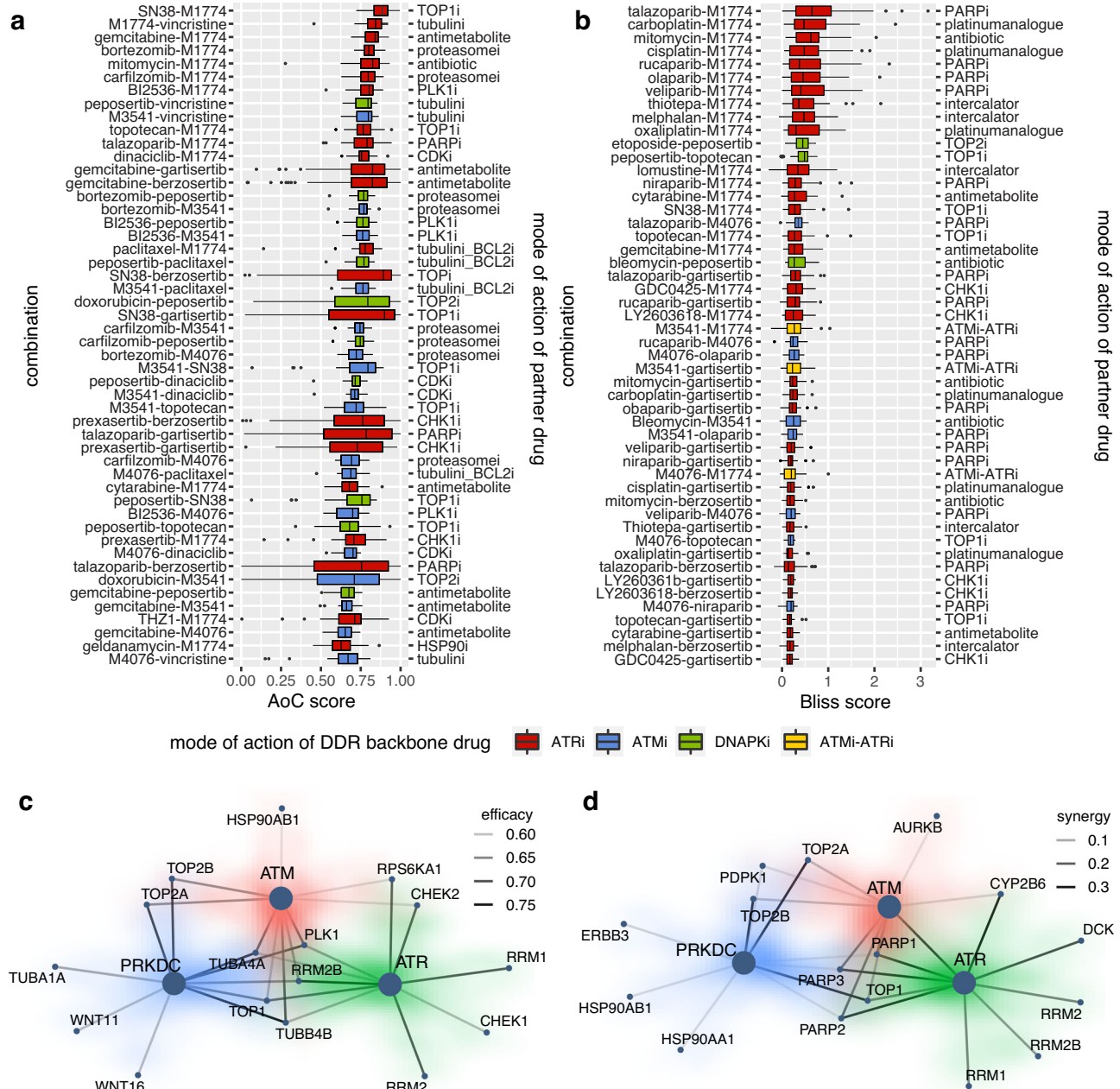

**Fig. 2 | Top DDR inhibitor combination treatments that achieve the highest efficacy and synergy across all cell lines in the high-throughput treatment screening in this study.** Boxplots showing the treatment responses of drug combinations with the top 50 averaged (**a**) efficacy and (**b**) synergy responses in all 62 cancer cell lines (*n* = 62). Drug combinations are shown on the left side. Mode-of-actions of the DDR inhibitors are denoted by red (ATR inhibitor), blue (ATM inhibitor), green (DNA-PK inhibitor), and yellow (ATR inhibitor-ATM inhibitor combination) in the box plot, while mode-of-actions of the partner drugs are shown at the right side. The interquartile range (25th to 75th percentile) and median lines are show, with whiskers extending to 1.5 times the interquartile range. **c**, **d** show the top 10 target genes with the highest average (**c**) efficacy and (**d**) synergy in combination with ATR, ATM, and DNA-PK (PRKDC) inhibitors. Each target gene of a partner drug is denoted by a node in the diagram, and the combination response (efficacy or synergy) is denoted by the relative strength of the connection.

information. In our data, both the sequencing as well as combination treatment response data were generated from the same cell culture lines, avoiding potential issues resulting from differing molecular backgrounds between screened and sequenced cell lines that may bias the analysis.

We identified inhibitors to four biological pathways that achieve strong combination efficacy in the screened cell lines when combined with any of the investigated DDR kinase inhibitors: the DNA topoisomerase pathway (TOP1 and TOP2 inhibitors), the serine/threonine-protein kinase PLK1 pathway (PLK1 inhibitors), the p53-inducible ribonucleotide reductase pathway (gemcitabine and cytarabine) and

cell cycle checkpoints (in particular, CHK1 inhibitors). In addition, we found that PARP inhibitors achieve strong synergistic effects in combination with the ATR and ATM inhibitors, a finding that is currently being investigated for ATRi in ongoing clinical trials[38,39].

Concerning drug combination synergy, we identified peposertib-gamma-ionizing-radiation (ionizing radiation) (DNA-PKi-IR) and M4076-berzosertib (ATMi-ATRi) as combination treatments that show cross-cancer type variability in efficacy and synergy. Peposertib-gamma-ionizing-radiation is a DDR inhibitor combination that has been actively under preclinical evaluation[22,40,41] and shows robust response in cervical cancer xenograft model[42] and enhances the

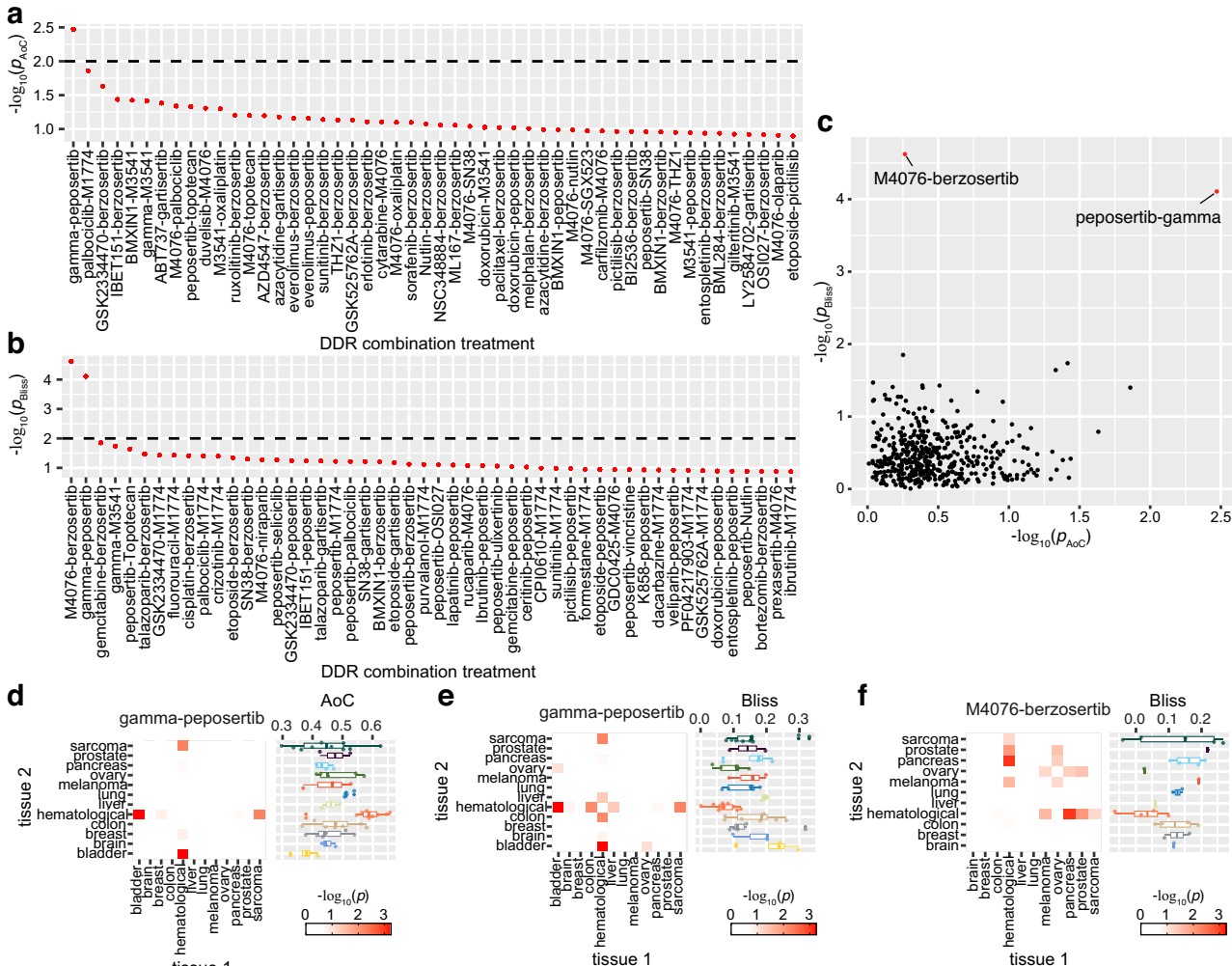

**Fig. 3 | Results from cross-cancer type variance test of DDR inhibitor combination treatment response. a, b** Kruskal−Wallis test shows the significance of cross-cancer type variance of DDR inhibitor combinations tested in this study. −log10(p) from the cross-tissue variance test for (**a**) efficacy (AoC score) and (**b**) synergy (Bliss) of the top 50 combinations are shown, and the significance threshold (p = 0.01) is marked by a dashed line. **c** shows the correlation between cross-cancer-type variance significance in AoC score and Bliss score for all combination treatments tested in this study. Each dot in (**c**) denotes a combination treatment. **d**−**f** Heatmap shows the results from post hoc analysis by Dunn's test on the significantly variant combination treatments (peposertib-gamma-ionizing-radiation and M4076-berzosertib) from the Kruskal−Wallis test, and the right lane shows the distribution of responses (AoC or Bliss scores) in different cancer types (boxplots show the 25, 50 and 75 percentiles with whiskers extending to 1.5 times the interquartile range; for each cancer types the total numbers of cell lines are: bladder = 4; brain = 3; breast = 6; colon = 8; hematological = 10; liver = 2; lung = 5; melanoma = 3; ovary = 5; pancreas = 4; prostate = 2; sarcoma = 10). As M4076-berzosertib only shows the cross-cancer-type variance in the Bliss score, only the post hoc test result on the Bliss score is shown for this combination. All statistically significant values from the variance test are two-sided.

response of immunotherapy[43]. Meanwhile, ATM and ATR loss-of-function have been proposed as being in a synthetically lethal relationship[44] and ATM has been identified as predictive biomarkers of single-agent ATRi in multiple tumor types[45–47]. Both combinations show synergy in vitro (0.14 bliss score for peposertib-gamma-ionizing-radiation combination and 0.11 bliss score for M4076-berzosertib combination), indicating the potential of further investigation of the proper indication of both combinations in clinical use.

Our investigation has yielded crucial evidence shedding light on the potential of DDR targeted combination therapies, highlighting their significant clinical prospects. However, it is essential for future studies to meticulously evaluate the toxicity and adverse events linked to such combined treatment approaches, ensuring patient safety and precise dosage calibration. The concept of synthetic lethality, which forms the foundation of DDR-targeted combination therapy, inherently enhances efficacy while concurrently increasing the risk of

toxicity and adverse events[48,49]. For example, PARP inhibitors, both as monotherapies and as components of combination regimens, have been extensively researched due to their pioneering role in DDR targeted therapy, with a clinical history spanning over a decade[50–53]. The simultaneous administration of the PARP inhibitor olaparib with the ATR inhibitor ceralasertib, for instance, has been correlated with the onset of anemia, neutropenia, and thrombocytopenia[54,55]. Furthermore, certain combinations elucidated in our current study have previously been reported to increase the incidence of toxicity and adverse events. The ATR inhibitor berzosertib, usually well-tolerated as a single-agent therapy, has shown an increased prevalence of adverse events and hematological toxicities, including anemia, nausea, and neutropenia, when combined with carboplatin[56], gemcitabine[57,58], or topotecan[59] in early-phase clinical trials. Despite the progress we have made in our research, we acknowledge that our efforts are still limited to the preliminary phase of in vitro high-throughput screening.

Therefore, a comprehensive exploration of in vivo toxicity associated with all the synergistic combinations unveiled in this study awaits future clinical trials.

## Methods

### Cell culture and drug response detection

This study is carried out on cell lines only and complies with all relevant ethical regulations of Merck Healthcare KGaA and the University of Michigan. All dose-response experiments were conducted at Oncolead GmbH & Co. KG (Karlsfeld, Germany). Cell lines were purchased directly from the ATCC, NCI, CLS and DSMZ cell line collections. The cell lines were grown in the media recommended by the suppliers in the presence of 100 U/ml penicillin G and 100 µg/ml streptomycin supplied with 10% FCS.

Cells were grown in a 5% $CO_2$ atmosphere. Cell growth and treatment were performed in 96-well microtiter plates CELLSTAR® (Greiner Bio-One, Germany). Cells harvested from exponential phase cultures by trypsinization or by splitting (in the case of suspension growing cells) were plated in 90 µl of media at optimal seeding densities. The optimal seeding density for each cell line was determined to ensure exponential growth for the duration of the experiment. All cells growing without anticancer agents were sub-confluent by the end of the treatment, as determined by visual inspection.

Cells were allowed to stay for another 48 h prior to compound treatment. The treatment was performed for 120 h and stopped by the addition of trichloroacetic acid followed by using a total protein staining protocol (Sulforhodamine B (SRB) staining)[60]. The bound SRB was solubilized with 100 µl of 10 mM Tris base. Optical density was measured at 492, 520, and 560 nm. Compound dilutions were performed in DMSO and diluted 1:100 in the RPMI medium. Combined treatment has been performed simultaneously. Ninety µl of cells were treated by mixing with 10 µl of the compound-containing media (resulting in a final DMSO concentration of 0.1%). In the case of combination, both agents were mixed together in DMSO at equal volumes so that the final concentration of DMSO was 0.2%. In addition, all experiments contained a few plates with cells that were analyzed immediately after the 48 h recovery period. These plates contained information about the cell number, $T_z$, at time zero, i.e., before treatment, and served to calculate the cytotoxicity.

The calculation nomenclature used was introduced by DTP of the NCI[61]. The first step in data processing was calculating an average background value for each plate, derived from plates and wells containing mediums without cells. The average background optical density was then subtracted from the appropriate control values (containing cells without the addition of a drug), from values representing the cells treated with an anticancer agent, and from values of wells containing cells at time zero. Thus, the following values were obtained for each experiment: control cell growth, C; cells in the presence of an anticancer agent $T_i$ and cells prior to compound treatment at time zero, $T_z$ (or $T_0$, in some publications).

The selection of the concentration range for all agents was based on previous experiments using a panel of 62 cell lines. A four-fold dilution and 5 data points were sufficient to cover the complete activity range for most of the agents (Supplementary Figs. 8 and 9).

### Dose-response evaluation measures

The non-linear curve fitting calculations were performed using algorithms and visualization tools using four-parameter log-logistic regression[62,63].

To obtain an estimate of treatment efficacy that encompasses both potency and maximum effect, the relative area over the curve (AoC) was computed by estimating the area under the fitted dose-response curve by the trapezoidal rule within ranges of relative growth rates compared to untreated controls between 0% and 100%, and within ranges of drug concentrations between 1 nM and 1 mM, and dividing the estimated area by the sum of areas below and above the curve. The relative AoC measure used in this work thus captures both the potency of a compound combination (usually measured by $IC_{50}$ or $GI_{50}$) as well as the maximum effect on cellular growth (as measured by the minimum of the curve); the relative AoC is of particular usefulness for capturing the efficacy of DDR inhibitors, many of which often have a comparatively low maximum effect less than 50% growth inhibition at realistic concentrations, which makes IC50 and GI50 less practically relevant.

Combination effects for the different compound combinations are calculated using the Bliss independence model[64,65] under the assumption of independent modes of action of the combination partners. Bliss excess was calculated as the average excess of the observed effect $E_{OBS}$ (i.e., the relative reduction of growth rate compared to untreated controls) over the calculated linear combination of the monotherapy treatments effects ($E_{1+2} = E_1 + E_2 - E_1 E_2$) for all concentrations used:

$$\text{Bliss}_{\text{excess}} = \frac{1}{n}\sum_{i=1}^{n} E_{\text{OBS}_i} - E_{1+2_i} \qquad (1)$$

In this formulation, the $\text{Bliss}_{\text{excess}}$ is a continuous value between −1 and 1 where values higher than about 0.2 are usually considered synergistic, and values below about −0.2 are usually considered antagonistic.

### Statistics and reproducibility

The reproducibility of measured response (i.e. AoC and Bliss score) are measured by Pearson's correlation within the replicated experiments. No data were excluded from the analyses.

### Quantification and statistical analysis for drug response variance test

For hierarchical clustering based on drug responses, we used heatmap.2 function of gplots module (3.1.3) from R (4.2.3) for hierarchical clustering using Euclidean as the distance function and ward.D2 as the cluster function.

We used Python (≥3.8) module *scipy* (1.11.3) to carry out the Kruskal–Wallis test to test if a drug has different responses between different cancer types. Kruskal–Wallis test is especially suitable for this situation as a non-parametric test, so it won't be affected by the different sample sizes of the subsets. For the significantly tissue-specific drugs ($p < 0.01$), we also used *scipy* to carry out post hoc tests, including Dunn's test, Mann–Whitney Pairwise test, Conover–Iman test and bootstrapping for 10,000 times to locate the significantly different tissue types. Bonferroni correction was performed to adjust the above multiple comparisons.

### Inclusion and ethics

This work adheres to the principles of inclusivity and ethical conduct. We have sought diverse perspectives, ensuring fair representation and acknowledging all contributors. Additionally, all underlying research was conducted ethically, with integrity, and in compliance with applicable guidelines and regulations.

### Reporting summary

Further information on research design is available in the Nature Portfolio Reporting Summary linked to this article.

## Data availability

The DDR combination in vitro screening data collected in this study are shared at and can be freely downloaded from: https://osf.io/8hbsx/. Source data are provided with this paper.

## Code availability

The source code of all statistical analyses is available from GitHub: https://github.com/GuanLab/DDR_combination_analysis.

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

## Acknowledgements

This study was sponsored by Merck (CrossRef Funder ID: 10.13039/100009945).

## Author contributions

Study design and data acquisition: S.-E.S., H.D., T.G., M.Z., and F.T.Z. Data analytics: H.Z. Manuscript writing: Y.G., H.Z., S.-E.S., and J.K. Figures: H.Z. All authors read and approved the manuscript.

## Competing interests

S.-E.S., H.D., J.K., T.G., and M.Z. are employees of Merck. F.T.Z. was employee of Merck at the time of study and no longer the employee of Merck at the time of publication. H.Z. reports no competing interests. Y.G. received research support from Merck during the period of this study.
