## [Peer Review File · Nature Communications]

REVIEWER COMMENTS

Reviewer #1 (Remarks to the Author): expertise in DNA damage response mechanisms

This manuscript describes the results of a study to test the combinatorial effects of DNA damage response kinase inhibitors (ATR, ATM and DNA-PK inhibitors) with other anti-cancer agents. A total of 17,912 combination treatments were examined. DNA topoisomerase inhibitors, PLK1 inhibitors, ribonucleotide reductase inhibitors, PARP inhibitors and cell cycle checkpoint inhibitors displayed the highest levels of synergy with ATM/ATR/DNA-PK inhibitors. Biomarkers associated with greater sensitivity to DNA-PK inhibitor/IR combination treatments and ATR inhibitor/ATM inhibitor combination treatments were identified.

This is a brute-force study intended to help the clinical development of ATR/ATM/DNA-PK inhibitors. The results are worth reporting. If not reported, I can imagine that many companies will have to repeat these findings, which is not the best use of resources.

Specific Comments

1. In terms of academic insights, the results reported in Figs 1 and 2 could form the basis for mechanistic studies. They could also point to the use of drug combinations in the clinic. However, many of these combinations have already been tested in the clinic and abandoned due to synthetic toxicity. It would make sense for the authors to stress that their study did not investigate the issue of synthetic toxicity, which for this class of drugs, affects the hematopoietic system. Therefore, these findings would be hard to translate to the clinic without further preclinical analysis in mouse or other models to explore potential bone marrow synthetic toxicities.
2. The correlations examining cancer types and identifying biomarkers are interesting and important, as they might help identify the patients that could benefit the most from the proposed combination treatments.
3. The Discussion needs to mention the limitations of this study and plans for further studies, as discussed in point 1 above. I would agree that it is beyond the scope of the current study to explore synthetic toxicities, but the potential for synthetic toxicities should be clearly stated. Also, if available, relevant phase I studies showing synthetic toxicities of the agents used here should be discussed.

Reviewer #2 (Remarks to the Author): expertise in drug screening methodologies

Zhang and colleagues report a large scale screen of drug combinations, tested against a diverse panel of cancer cell cultures, with a focus on inhibitors of DNA damage response. This academic-industry collaboration has supported a comprehensive scale of experimentation and exhaustive 'omics' profiling of the many cancer cell lines studied for the purpose of biomarker analyses. I anticipate that these data will be interesting to the field of cancer combination therapy, if they are published in a manner that could support other analyses. The data currently available online (DOI 10.17605/OSF.IO/8HBSX) has more of a 'summary statistic' nature, and I anticipate that some readers would appreciate the underlying dose-response data. For example, some people may wish to analyze drug-drug interactions via different models, or much could be learned by analyzing which pairs of mechanisms correspond to which shapes of dose-response surfaces (as described in PMID: 17332758). Such analyses could be supported by releasing dose-response matrices, but cannot be done only with AOC and Bliss scores as summary metrics. Given the large scale of this study, I do not think that in vivo experiments should be seen as necessary or feasible; especially given the 'pan-cancer' set of model systems used here, I see animal studies as belonging to the

scope of future research.

The major result of the study is the identification of several co-inhibitory strategies showing efficacy and synergy with ATM/ATR/DNA-PK inhibition across a variety of cell cultures. These findings are robustly supported, because the use of many drugs and many cell cultures leads to the identification of certain mechanistic clusters where combinations of certain mechanisms (represented by independent experiments of specific drugs) are consistently effective and/or synergistic. These results are not all that well illustrated though, with many of the main figures having more of a schematic nature without showing dose-response surfaces for the most interesting drug combinations (this isn't to say I doubt the data, it is just isn't shown much and one needs to download the supporting data to see the key findings). The level of experimental reproducibility compares very favorably to the most robust public datasets on drug responses in cancer cell cultures. I very much appreciate that the article reported both efficacy and synergy as distinct metrics.

I have a question about quantifying efficacy, since AOC depends on the concentrations chosen. If a drug was studied at concentrations higher (or lower) than the clinically relevant dose, it may appear to have high (or low) efficacy as a result of the concentration range studied. Given this influence, it is not clear whether combinations with higher AOC are more likely to be clinically effective than combinations with lower AOC, but this question might be answered, to the best ability of pre-clinical systems, by analyzing efficacy at the points of the dose-response matrices that most closely correspond to the clinically relevant doses of these drugs. In other words, I think you could get a more clinically relevant metric from a re-analysis of the current data. Clinically relevant concentrations of many cancer therapies can be found in these articles: PMID 33203866, PMID: 28364015. If the authors would rather leave this for future work, it is another reason to publish granular dose-response data.

The article states that "ideal drug combinations are not only safe and effective, but also complement each other in a synergistic manner". This notion is commonly stated but is not supported by clinical evidence nor this dataset, which shows a low correlation between efficacy and synergy. Synergy is rarely exhibited by approved drug combinations used to treat humans with cancer, and because it is not synonymous with efficacy, we must remember it does not necessarily mean more effective. Synergy, as formally defined in this article and others, measures whether the efficacy is expected or unexpected compared with the constituent drugs' activities. Patient outcome depends on safety and efficacy, and does not depend on whether a molecular biologist is or isn't surprised by the efficacy. This doesn't mean drug interactions aren't interesting to identify and understand, but it emphasizes the value of measuring efficacy at a clinically relevant dose. A recent analysis (PMID: 33203866) comparing pre-clinical and clinical trial results suggests that tumor cell inhibition at the clinically relevant dose is a good predictor of clinical efficacy, which shouldn't come as a surprise, and which has implications for the analysis and interpretation of the new data in this study.

The second half of the paper focuses on molecular features that correlate with synergy. These analyses have not demonstrated rigor with respect to a large amount of multiple hypothesis testing. Please consider: if you were to scramble the drug response labels of the cell cultures, and repeat the effort to detect biomarkers in a dataset that was actually just noise (because of scrambling the labels), you would still detect lots of highly significant correlations. Some analysis is needed to assess whether the observed correlations are any more robust than what you would find in noise. Any of the following three suggestions might fill this need: (1) You could use a process of repeatedly scrambling the data labels, performing the 'biomarker discovery', and seeing what strengths of biomarkers are found in noise. Doing this repeatedly would quantify false discovery rate for biomarkers (correlative features) of various strengths. For example, you might find that a correlation of 0.5 has an FDR of 10%, while a correlation of 0.3 has an FDR of 50% (which would mean it is easy to find correlations of 0.3 in noise). An objective FDR cut-off could be applied to define what results should be trusted. (2) You could divide the large data set into 'training' and 'test' sets, and thereby quantify predictive performance of possible biomarkers. (3) Given that this search for correlative features is hypothesis-generating, not confirmatory, you could analyze external datasets for independent validation. While this could not be applied to novel combinations studied for the first time here, biomarkers of single-drug response (as in Supplementary Figures S6 to S9) could be tested for validation in datasets like the Broad

Institute's CTRP (which also has drug response data and 'omics' of cell cultures available for download).

While some biomarker associations may be real, given that correlations are being sought with thousands of possible genes, and thousands of possible gene sets, it is likely that most of the reported associations are spurious and will not replicate in independent data. Overall this portion of the article could be either omitted entirely, or replaced with a more rigorous approach to testing biomarker performance. Also, given the above notes on the clinical relevance of 'synergy' versus 'efficacy', it is unclear whether biomarkers of synergy are more useful than biomarkers of efficacy – but these data could also support a search for biomarkers of efficacy.

The methods section is detailed and clear. One vital detail that should be added (eg. in a supplementary table) is seeding density used for each particular cell culture. As the authors recognize, seeding density has a critical influence on these experiments, and without this information, these experiments would not be reproducible by others. Another vital detail is the drug concentrations studied. The currently available data lists drug concentrations in some rows but not all, and without this information, these experiments would not be reproducible by others. It would also be helpful to demonstrate (eg. in a supplementary figure) some actual dose-response surfaces or heatmaps, and their replicates.

Some errors in the data visualization are present in Supplementary Figures. Figures S3, S4, and S5 all show large blocks in which an identical level of efficacy, or synergy, is present across an entire row. As different columns represent different cell lines, this suggests that around 30 cell cultures all had perfectly identical drug responses to one another, for every drug combination tested. This is of course not believable and must reflect some error in the figures. I have examined the underlying data and fortunately this error is not present in the data; it must only be an issue with the figures.

-Adam Palmer

Reviewer #3 (Remarks to the Author): expertise in bioinformatics and machine learning

Review of a manuscript entitled "Mapping combinatorial drug effects to DNA damage response kinase inhibitors"

The article describes a study on the combination of inhibitors targeting DNA damage response (DDR) kinases ATR, ATM, and DNA-PK with anti-cancer agents. The study identified five DDR-related pathways that displayed high combinatorial efficacy and synergy with these inhibitors, and potential molecular biomarkers associated with treatment response was also identified through genomic and transcriptomic profiling.

I have some comments, concerns, and clarifications that need to be addressed by the authors. To properly evaluate the work, it is necessary to first clarify whether the results shown were obtained by correctly applying appropriate statistical and methodological approaches. Therefore, I will be formulating different questions to ensure this is the case and subsequently evaluate whether the conclusions are appropriate. It is essential to establish that the data has been appropriately preprocessed, normalized, and quality controlled. Moreover, it is crucial to confirm that the statistical tests employed are suitable for the type of data and hypotheses being tested. Additionally, we should examine whether the data distribution assumptions have been appropriately checked and that the potential confounding factors have been adequately addressed. Only by ensuring these aspects, we can evaluate whether the results obtained support the conclusions drawn by the authors.

1) Using post-hoc analysis to adjust hypothesis through Dunn's test is not recommended because it is known to have the lowest statistical power among the options available.

2) What is the justification for using the hg19 reference genome?

3) The `knn.impute` approach is a straightforward method for imputing missing values, but it may not be suitable for all datasets, particularly in the case of sequencing data. There are several advanced imputation techniques that can be explored to handle missing values in such data, such as multiple imputation, Bayesian imputation, and matrix factorization-based imputation methods. These techniques can handle complex relationships and correlations within the data, which can be particularly useful in high-dimensional sequencing datasets where multiple variables may be interdependent. Therefore, while the `knn.impute` method may be useful in some cases, it is essential to explore more advanced approaches to handle missing values in sequencing data.

4) It is essential to provide clear and detailed descriptions of the methods used in a research study, to enable reproducibility and facilitate the assessment of the validity of the results. While using a pre-existing workflow such as `bcio` can be a useful tool for data processing and analysis, it may not provide sufficient information for readers who are not familiar with the workflow. It is therefore crucial to supplement the use of such workflows with detailed explanations of the steps taken and the parameters used, to ensure that readers can understand and reproduce the methods. Without clear and detailed descriptions, readers may not be able to assess the validity of the methods or replicate the results, which can undermine the credibility of the study. Therefore, it is important to provide a clear and thorough explanation of the methods used, regardless of the tools or workflows employed.

5) The use of hypergeometric tests is a widely used approach to evaluate gene sets for enrichment analysis. While this method can be effective, it is important to consider that there are other options available for evaluating gene sets that may provide additional insights. For example, gene set enrichment analysis (GSEA) evaluates whether a set of genes shows significant differences in expression levels between two or more biological conditions, and can be useful for identifying pathways and biological processes that are differentially regulated.

6) What is the rationale for using the Kruskal-Wallis test, has its applicability been checked according to the conditions it should meet, and what multiple testing correction has been applied?

7) It has been reported that Spearman's correlation coefficient was used in the analysis, which is appropriate if it has been checked that the relationship between the variables is not linear or if the data does not meet the assumptions for Pearson's correlation coefficient. Although Spearman's test is more robust against outliers, it is important to ensure that its use is justified and that the underlying assumptions are met.

8) What was the curating process used to obtain the list of 2725 genes?

It seems that sometimes there is not enough information to adequately follow the work carried out, making it difficult to fully understand the magnitude of the study. In order to properly evaluate the results and conclusions, it is important to have a clear understanding of the methods and analyses employed, as well as the limitations and potential biases that may exist. Additional details and explanations could greatly enhance the clarity and transparency of the research, and would enable readers to more effectively assess the findings and potential implications.

RESPONSE TO REVIEWERS' COMMENTS

All reviewer's comments are marked in black

All responses are marked in blue.

Reviewer #1 (Remarks to the Author): expertise in DNA damage response mechanisms

This manuscript describes the results of a study to test the combinatorial effects of DNA damage response kinase inhibitors (ATR, ATM and DNA-PK inhibitors) with other anti-cancer agents. A total of 17,912 combination treatments were examined. DNA topoisomerase inhibitors, PLK1 inhibitors, ribonucleotide reductase inhibitors, PARP inhibitors and cell cycle checkpoint inhibitors displayed the highest levels of synergy with ATM/ATR/DNA-PK inhibitors. Biomarkers associated with greater sensitivity to DNA-PK inhibitor/IR combination treatments and ATR inhibitor/ATM inhibitor combination treatments were identified.

This is a brute-force study intended to help the clinical development of ATR/ATM/DNA-PK inhibitors. The results are worth reporting. If not reported, I can imagine that many companies will have to repeat these findings, which is not the best use of resources.

Specific Comments

1. In terms of academic insights, the results reported in Figs 1 and 2 could form the basis for mechanistic studies. They could also point to the use of drug combinations in the clinic. However, many of these combinations have already been tested in the clinic and abandoned due to synthetic toxicity. It would make sense for the authors to stress that their study did not investigate the issue of synthetic toxicity, which for this class of drugs, affects the hematopoietic system. Therefore, these findings would be hard to translate to the clinic without further preclinical analysis in mouse or other models to explore potential bone marrow synthetic toxicities.

We appreciate the reviewer for the insightful comments on synthetic toxicity. It's true that the DDR targeted combination therapy has been faced with the risk of increased "synthetic" toxicity in clinics. As DDR-targeted combination therapy could induce synthetic lethality, there are many cases that this kind of usage can also increase the toxicity to hematological systems and adverse events^{1,2}. For example, the PARP inhibitor Olaparib has been found to induce adversarial events such as anemia, neutropenia and thrombocytopenia when used in combination with ATR inhibitor Ceralasertib^{3,4}. The adverse events of PARP inhibitors alone and in combination therapies has been comprehensively studied and extensively discussed in the past, as it's the first successful DDR targeted therapeutic agents and has been used in clinics for more than a decade⁵⁻⁸.

On the other hand, the adverse events of other DDR targeted drugs, such as ATR, ATM or DNA PK inhibitors, which are the main DDR inhibitors we studied in this paper, are less discussed. We still found many of the combinations we discussed in this paper have been discovered with toxicity and adverse events in previous studies. For example, the ATR inhibitor Berzosertib, while is well tolerant by patients when used as a monotherapy, has induced increased adverse events and hematological toxicities such as anemia, nausea and neutropenia, when used in combination with carboplatin⁹, gemcitabine^{10,11}, topotecan¹², in phase I or phase II clinical studies. However, as our study was still limited at an early stage of *in vitro* high-throughput screening, the *in vivo* toxicity of all the efficacious/synergistic combinations we found in this study still needs extensive evaluations in following clinical trials. We agree with and appreciate the reviewer's comment and decide to discuss this as one of the limitations of our study in the discussion section.

According to the above reasons, we add a discussion section on synthetic toxicities.

“Our investigation has yielded crucial evidence shedding light on the potential of DDR targeted combination therapies, highlighting their significant clinical prospects. However, it is essential for future studies to meticulously evaluate the toxicity and adverse events linked to such combined treatment approaches, ensuring patient safety and precise dosage calibration. The concept of synthetic lethality, which forms the foundation of DDR-targeted combination therapy, inherently enhances efficacy while concurrently increasing the risk of toxicity and adverse events^{1,2}. For example, PARP inhibitors, both as monotherapies and as components of combination regimens, have been extensively researched due to their pioneering role in DDR targeted therapy, with a clinical history spanning over a decade⁵⁻⁸. The simultaneous administration of the PARP inhibitor Olaparib with the ATR inhibitor Ceralasertib, for instance, has been correlated with the onset of anemia, neutropenia, and thrombocytopenia^{3,4}. Furthermore, certain combinations elucidated in our current study have previously been reported to increase the incidence of toxicity and adverse events. The ATR inhibitor Berzosertib, usually well-tolerated as a single-agent therapy, has shown an increased prevalence of adverse events and hematological toxicities, including anemia, nausea, and neutropenia, when combined with carboplatin⁹, gemcitabine^{10,11}, or topotecan¹² in early-phase clinical trials. Despite the progress we have made in our research, we acknowledge that our efforts are still limited to the preliminary phase of in vitro high-throughput screening. Therefore, a comprehensive exploration of in vivo toxicity associated with all the synergistic combinations unveiled in this study awaits future clinical trials.”

2. The correlations examining cancer types and identifying biomarkers are interesting and important, as they might help identify the patients that could benefit the most from the proposed combination treatments.

We appreciate your comments on the identification of biomarkers on cancer types. We agree that these biomarkers provide a potential for identifying patients that are most suitable for DDR targeted combination treatments.

3. The Discussion needs to mention the limitations of this study and plans for further studies, as discussed in point 1 above. I would agree that it is beyond the scope of the current study to explore synthetic toxicities, but the potential for synthetic toxicities should be clearly stated. Also, if available, relevant phase I studies showing synthetic toxicities of the agents used here should be discussed.

We agree with and appreciate the reviewer’s suggestion. Please refer to our response and changes made in comment (1).

Reviewer #2 (Remarks to the Author): expertise in drug screening methodologies

Zhang and colleagues report a large scale screen of drug combinations, tested against a diverse panel of cancer cell cultures, with a focus on inhibitors of DNA damage response. This academic-industry collaboration has supported a comprehensive scale of experimentation and exhaustive ‘omics’ profiling of the many cancer cell lines studied for the purpose of biomarker analyses. I anticipate that these data will be interesting to the field of cancer combination therapy, if they are published in a manner that could support other analyses. The data currently available online (DOI 10.17605/OSF.IO/8HBSX) has more of a ‘summary statistic’ nature, and I anticipate that some readers would appreciate the underlying dose-response data. For example, some people may wish to analyze drug-drug interactions via different models, or much could be learned by analyzing which pairs of mechanisms correspond to which shapes of dose-response surfaces (as described in PMID: 17332758). Such analyses could be supported by releasing dose-response matrices, but cannot be done only with AOC and Bliss scores as summary metrics. Given the large scale of this study, I do not think that in vivo experiments should be seen as necessary or feasible; especially

given the ‘pan-cancer’ set of model systems used here, I see animal studies as belonging to the scope of future research.

The major result of the study is the identification of several co-inhibitory strategies showing efficacy and synergy with ATM/ATR/DNA-PK inhibition across a variety of cell cultures. These findings are robustly supported, because the use of many drugs and many cell cultures leads to the identification of certain mechanistic clusters where combinations of certain mechanisms (represented by independent experiments of specific drugs) are consistently effective and/or synergistic. These results are not all that well illustrated though, with many of the main figures having more of a schematic nature without showing dose-response surfaces for the most interesting drug combinations (this isn’t to say I doubt the data, it is just isn’t shown much and one needs to download the supporting data to see the key findings). The level of experimental reproducibility compares very favorably to the most robust public datasets on drug responses in cancer cell cultures. I very much appreciate that the article reported both efficacy and synergy as distinct metrics.

1. I have a question about quantifying efficacy, since AOC depends on the concentrations chosen. If a drug was studied at concentrations higher (or lower) than the clinically relevant dose, it may appear to have high (or low) efficacy as a result of the concentration range studied. Given this influence, it is not clear whether combinations with higher AOC are more likely to be clinically effective than combinations with lower AOC, but this question might be answered, to the best ability of pre-clinical systems, by analyzing efficacy at the points of the dose-response matrices that most closely correspond to the clinically relevant doses of these drugs. In other words, I think you could get a more clinically relevant metric from a re-analysis of the current data. Clinically relevant concentrations of many cancer therapies can be found in these articles: PMID 33203866, PMID: 28364015. If the authors would rather leave this for future work, it is another reason to publish granular dose-response data.

We appreciate the reviewer for his questions and suggestions. The concentrations that the drugs have been measured at is truly a vital factor in analyzing efficacy, and truly leaves a lot of space for future works.

We have uploaded the granular dose-response data to <https://osf.io/8hbsx/>, as the reviewer suggested. We hope this data will be valuable for building future studies on DDR targeted combination therapy.

2. The article states that “ideal drug combinations are not only safe and effective, but also complement each other in a synergistic manner”. This notion is commonly stated but is not supported by clinical evidence nor this dataset, which shows a low correlation between efficacy and synergy. Synergy is rarely exhibited by approved drug combinations used to treat humans with cancer, and because it is not synonymous with efficacy, we must remember it does not necessarily mean more effective. Synergy, as formally defined in this article and others, measures whether the efficacy is expected or unexpected compared with the constituent drugs’ activities. Patient outcome depends on safety and efficacy, and does not depend on whether a molecular biologist is or isn’t surprised by the efficacy. This doesn’t mean drug interactions aren’t interesting to identify and understand, but it emphasizes the value of measuring efficacy at a clinically relevant dose. A recent analysis (PMID: 33203866) comparing pre-clinical and clinical trial results suggests that tumor cell inhibition at the clinically relevant dose is a good predictor of clinical efficacy, which shouldn’t come as a surprise, and which has implications for the analysis and interpretation of the new data in this study.

We thank the reviewer for his comments. It is true that the correlation between efficacy and synergy is low. Based on our own dataset, Pearson’s correlation between AoC and Bliss is 0.2 ($p < 1e-22$) (Page 3). When we stated “ideal drug combinations are not only safe and effective, but also complement each other in a synergistic manner”, we wanted to express that the “favorable” drug combination should be both efficacious and synergistic, while co-occurrence of efficacy and synergy is not common.

However, the insight that the reviewer provided is very helpful. It is true that the treatments' effectiveness is as expected and can be precisely controlled is more meaningful in clinical practice. As we provided the full dose response matrix with treatment response at a certain dose, this data can be helpful in answering this question by comparing this data to clinical effective dosage.

3. The second half of the paper focuses on molecular features that correlate with synergy. These analyses have not demonstrated rigor with respect to a large amount of multiple hypothesis testing. Please consider: if you were to scramble the drug response labels of the cell cultures, and repeat the effort to detect biomarkers in a dataset that was actually just noise (because of scrambling the labels), you would still detect lots of highly significant correlations. Some analysis is needed to assess whether the observed correlations are any more robust than what you would find in noise. Any of the following three suggestions might fill this need:

(1) You could use a process of repeatedly scrambling the data labels, performing the 'biomarker discovery', and seeing what strengths of biomarkers are found in noise. Doing this repeatedly would quantify false discovery rate for biomarkers (correlative features) of various strengths. For example, you might find that a correlation of 0.5 has an FDR of 10%, while a correlation of 0.3 has an FDR of 50% (which would mean it is easy to find correlations of 0.3 in noise). An objective FDR cut-off could be applied to define what results should be trusted.

(2) You could divide the large data set into 'training' and 'test' sets, and thereby quantify predictive performance of possible biomarkers.

(3) Given that this search for correlative features is hypothesis-generating, not confirmatory, you could analyze external datasets for independent validation. While this could not be applied to novel combinations studied for the first time here, biomarkers of single-drug response (as in Supplementary Figures S6 to S9) could be tested for validation in datasets like the Broad Institute's CTRP (which also has drug response data and 'omics' of cell cultures available for download).

While some biomarker associations may be real, given that correlations are being sought with thousands of possible genes, and thousands of possible gene sets, it is likely that most of the reported association are spurious and will not replicate in independent data. Overall this portion of the article could be either omitted entirely, or replaced with a more rigorous approach to testing biomarker performance. Also, given the above notes on the clinical relevance of 'synergy' versus 'efficacy', it is unclear whether biomarkers of synergy are more useful than biomarkers of efficacy – but these data could also support a search for biomarkers of efficacy.

We thank the reviewer for this great suggestion. The idea of using resampling to improve the strength for biomarker discovery is indeed a good idea. We decided to take the first suggestion of simulating the "biomarker discovery" process to increase the power of our findings.

In Figure 4, we discussed the biomarkers for the two combinations we found that has significant variability between different cancer types: gamma-peposertib and M4076-Berzosertib; both combinations were tested against 62 cell lines from 12 different tissues.

For each of the treatments, We scrambled their response experiment by subsampling 80% of all samples, and then rediscover the biomarkers for the response by correlation analysis. This process is repeated one thousand times. From this repeated biomarker discovery process, we can compute the power for each of the biomarker's associations and their empirical FDR.

The figure below shows an example of the results we generated from the one-thousand scrambling of M4076-Berzosertib's response (Bliss score). All biomarkers (gene expression level) were ordered by their average association with the treatment response. The associations (Pearson's correlation coefficient) were colored as blue if they are positive and red if negative. This graph straightforwardly shows the biomarkers in the middle of this graph can be positively and negatively correlated with the response in different

subsampling rounds. We see these biomarkers as noise. On the far right and left side of this graph, where the biomarkers are consistently positive or negative correlated with the response in all scrambling rounds, they have an empirical False Discovery Rate (FDR) lower than 0.1%. In this case, the biomarkers at both sides (in red blocks) as a high significance of less than 0.1 FDR.

At the same time, we can also generate volcano plots for the biomarkers (figure below). By defining the cutoff of FDR and Pearson's correlation, we can obtain a group of highly confident biomarkers that are both highly associated with treatment response and highly confident.

Based on the method above, we checked the results we present in **Figure 4C and D**. Interestingly, for all biomarkers mentioned in **Figure 4**, their FDR is less than 0.01%. Therefore we made the following changes to our manuscript:

Fig. 4. Top biomarkers with the top association of combination treatment response of peposertib-gamma-ionizing-radiation and M4076-berzosertib. (A and B) demonstrate the biomarker discovery process to generate highly confident biomarkers. (A) In each biomarker discovery round, we scrambled the response experiment by subsampling 80% of all samples, and then rediscovered the biomarkers for the response by correlation analysis. (B) shows the distribution of all biomarker's correlation with the drug response after repeating the biomarker discovery process for 1,000 times. On the left panel, all biomarkers (gene expression level) were ordered by their average association with the treatment response. The associations (Pearson's correlation coefficient) were colored as blue if they are positive and red if negative. The biomarkers that are consistently positively or negatively correlated with the response have a lower than

0.1% FDR. The right panel presents a volcano plot showing the relationship between empirical FDR during the 1000 repeats and correlation between response and the biomarker (gene expression level). (C and D) represent top biomarkers of highest association with synergy (Bliss score) of (C) peposertib-gamma-ionizing-radiation and (D) M4076-berzosertib, respectively. In each subplot, single gene biomarkers in terms of expression level, loss-of-function, copy number variation, single nucleotide variation, and gene set biomarkers for expression patterns and loss-of-function patterns are shown. The association of biomarkers was evaluated by Pearson's correlation coefficient between its magnitude and the corresponding Bliss scores of the combination treatment on the cell lines. The direction of correlation is denoted by red (negative) and blue (positive). Biomarkers were ranked by significance ($-\log(p\text{-value})$) of Pearson's correlation coefficient.

We updated the Supplementary tables of TableS10 to TableS12 with our bootstrapped FDR showing statistical significance of all 10461 biomarkers tested in this study and added Figure S10-S13 with the biomarker rediscovery analysis results.

Figure S10. Diagram showing empirical FDR for biomarker discovery for peposertib-gamma-ionizing-radiation combination efficacy. Correlations between all types of biomarkers (exp, cnv, snv, lof, coh_pat and lof_pat) and combination treatment response of all biomarkers during bootstrapping of 1,000 times. Figure S10-S11 are complete results of Figure 4 B left panel, which shows the FDR estimation for the biomarker discovery process.

Figure S11. Diagram showing empirical FDR for biomarker discovery for peposertib-gamma-ionizing-radiation combination synergy. Correlations between all types of biomarkers (exp, cnv, snv, lof, coh_pat and lof_pat) and combination treatment response of all biomarkers during bootstrapping of 1,000 times.

Figure S12. Diagram showing empirical FDR for biomarker discovery for M4076-berzosertib combination synergy. Correlations between all types of biomarkers (exp, cnv, snv, lof, coh_pat and lof_pat) and combination treatment response of all biomarkers during bootstrapping of 1,000 times.

M4076-berzosertib

Figure S13. Volcano plots showing the relationship between empirical FDR and biomarkers' correlation with treatment response. This figure shows the complete results of Figure 4 B right panel.

4. The methods section is detailed and clear. One vital detail that should be added (eg. in a supplementary table) is seeding density used for each particular cell culture. As the authors recognize, seeding density has a critical influence on these experiments, and without this information, these experiments would not be reproducible by others. Another vital detail is the drug concentrations studied. The currently available data lists drug concentrations in some rows but not all, and without this information, these experiments would not be reproducible by others. It would also be helpful to demonstrate (eg. in a supplementary figure) some actual dose-response surfaces or heatmaps, and their replicates.

We thank the reviewer for his suggestions. Here we provided the details for the above information.

1. Seeding density: the cell culture and seeding density information is in the Methods section:

“All dose-response experiments were conducted at Oncolead GmbH & Co. KG (Karlsfeld, Germany). Cell lines were purchased directly from the ATCC, NCI, CLS and DSMZ cell line collections. The cell lines were grown in the media recommended by the suppliers in the presence of 100 U/ml penicillinG and 100 µg/ml streptomycin supplied with 10% FCS.

Cells were grown in a 5% CO₂ atmosphere. Cell growth and treatment were performed in 96-well microtiter plates CELLSTAR® (Greiner Bio-One, Germany). Cells harvested from exponential phase cultures by trypsinization or by splitting (in the case of suspension growing cells) were plated in 90 µl of media at optimal seeding densities. The optimal seeding density for each cell line was determined to ensure

exponential growth for the duration of the experiment. All cells growing without anticancer agents were sub-confluent by the end of the treatment, as determined by visual inspection.” (page 8)

The exact specifications of the seeding density were done at the vendor site and are part of their customized lab process, so we don't have the exact number. However the cells were seeded at a level that is individually optimal for each cell line to stay at exponential growth rate for the duration of the experiment, and we think this information should be sufficient.

2. Concentrations studied:

We uploaded the dose-response data to <https://osf.io/8hbsx/>. The responses are measured using different dose matrices (1×6 , 1×7 , 2×6 , 2×7 , 2×8 , 3×7 , 5×8 , 6×7 , 6×19 , 7×7 , 7×12 , 7×19 , 8×8 , 11×11 , 11×19 , 12×12 . The dose ranges are determined by the empirical effective range of the corresponding drugs. Also, we plotted the actual dose-response surfaces of the two combinations of interests in our study, peposertib-gamma-ionizing-radiation and M4076-berzosertib in Figure S14 and S15.

Figure S14. Demonstration of the dose-response matrices of peposertib-gamma-ionizing-radiation combination treatment. The responses (growth inhibition rate, GI) in cell lines at different doses of peposertib (mol) and gamma ionizing-radiation (Gy) were shown by heatmaps.

Figure S15. Demonstration of the dose-response matrices of M4076-berzosertib combination treatment. The responses (growth inhibition rate, GI) in cell lines at different doses of M4076 (mol) and berzosertib (mol) were shown by heatmaps.

5. Some errors in the data visualization are present in Supplementary Figures. Figures S3, S4, and S5 all show large blocks in which an identical level of efficacy, or synergy, is present across an entire row. As different columns represent different cell lines, this suggests that around 30 cell cultures all had perfectly identical drug responses to one another, for every drug combination tested. This is of course not believable and must reflect some error in the figures. I have examined the underlying data and fortunately this error is not present in the data; it must only be an issue with the figures.

We thank the reviewers for this insightful comment. It is true this figure has some problems. Since not all drug combinations were tested on all 62 cell lines, there are some missing values, and we used KNN to impute these missing values and it is a coarse way. However, we realized the KNN imputation will impede the true data presentation. As none of the downstream analysis was based on the imputed dataset, we decided removing the KNN imputation could be better. Therefore we removed the KNN impute process and presented the original data in Figure S3, S4 and S5.

-Adam Palmer

Reviewer #3 (Remarks to the Author): expertise in bioinformatics and machine learning

Review of a manuscript entitled "Mapping combinatorial drug effects to DNA damage response kinase inhibitors"

The article describes a study on the combination of inhibitors targeting DNA damage response (DDR) kinases ATR, ATM, and DNA-PK with anti-cancer agents. The study identified five DDR-related pathways that displayed high combinatorial efficacy and synergy with these inhibitors, and potential molecular biomarkers associated with treatment response was also identified through genomic and transcriptomic profiling.

I have some comments, concerns, and clarifications that need to be addressed by the authors. To properly evaluate the work, it is necessary to first clarify whether the results shown were obtained by correctly applying appropriate statistical and methodological approaches. Therefore, I will be formulating different

questions to ensure this is the case and subsequently evaluate whether the conclusions are appropriate. It is essential to establish that the data has been appropriately preprocessed, normalized, and quality controlled. Moreover, it is crucial to confirm that the statistical tests employed are suitable for the type of data and hypotheses being tested. Additionally, we should examine whether the data distribution assumptions have been appropriately checked and that the potential confounding factors have been adequately addressed. Only by ensuring these aspects, we can evaluate whether the results obtained support the conclusions drawn by the authors.

1) Using post-hoc analysis to adjust hypothesis through Dunn's test is not recommended because it is known to have the lowest statistical power among the options available.

We thank the reviewer for his suggestions. We understand that using Dunn's test may be questioned due to its potentially lower power compared to tests that assume equal group sizes and distributional assumptions. In our case, the multiple comparison test is carried out for groups with different sample sizes and distributions. The multiple comparisons are carried out between different cancer types, and each cancer type includes different numbers of cancer cell lines, in which many more powerful post-hoc tests, such as Tukey's Honestly Significant Difference (HSD) test, are not applicable. Dunn's test, while is more conservative and less likely to reject the null hypothesis when it's false, is more likely to avoid the type I errors in our reports.

We therefore, compared Dunn's test's results to other widely used post-hoc tests that are applicable for unequal variances and sample sizes. Besides Dunn's test, we also carried out Conover's test, Mann-whitney pairwise tests with Bonferroni correction, and bootstrapping.

As we do not know if the two samples have the same distribution we compared the the average of two samples in two-sample t-test. If the t-test is significant (i.e. $t > 1.96$), then the two groups are different, else not. The bootstrapped p-value is counted as the probability in the 10,000 boosted rounds that the two samples compared are significantly different.

Below shows the results of all post-hoc analysis results on gamma-peposertib combination efficacy. The other tests (Conover, Mann-whitney and bootstrapping) showed the same results as Dunn's test. It shows the findings that we got from the Dunn's test are supported by other tests as well.

We thank the reviewer for bringing up the question for using Dunn's test in post-hoc analysis. As a reply to your question, we put the above results to the supplementary results to further support our findings:

Fig. S2. Heatmap shows the results from post-hoc analysis on the significantly variant monotherapy and combination treatments from the Kruskal-Wallis test, and the right lane shows the distribution of response score (AoC or Bliss) in different cancer types. (A) shows post-hoc analysis results of combination therapy peposertib-gamma-ionizing-radiation and M4076-berzosertib, and (B) shows post-hoc analysis results of monotherapy doxorubicin, M3541, peposertib, and oxaliplatin, respectively.

2) What is the justification for using the hg19 reference genome?

We thank the reviewer for bringing up this question. It is truly necessary to require a justification of using the hg19 reference genome.

However, as it has required years to obtain our DDR combination response screening data, when we carried out the whole genome sequencing of the cell lines in our project, the hg38 genome was not released yet. We will update the reference genome in the future.

3) The `knn.impute` approach is a straightforward method for imputing missing values, but it may not be suitable for all datasets, particularly in the case of sequencing data. There are several advanced imputation techniques that can be explored to handle missing values in such data, such as multiple imputation, Bayesian imputation, and matrix factorization-based imputation methods. These techniques can handle complex relationships and correlations within the data, which can be particularly useful in high-dimensional sequencing datasets where multiple variables may be interdependent. Therefore, while the `knn.impute` method may be useful in some cases, it is essential to explore more advanced approaches to handle missing values in sequencing data.

We thank the reviewer for his question on the KNN imputation. The reviewer 2 also brought up a related question on the demonstration of our dataset. We agree that the KNN imputation does not align well with our goal of providing an accurate representation of the dataset in this study.

As we are primarily focused on publishing the first original DDR-targeted combination therapy screening dataset, and the KNN imputation was only used for figure display and there is no subsequent analysis based on the imputed results, we have decided to remove the imputation when representing our data. We believe that showing the original dataset is of greater priority in this study, as it is the first large-scale DDR-targeted high-throughput screening dataset that is publicly available and has more scientific value. The more complex imputation methods that the reviewer has suggested, while more advanced, also rely on a large number of assumptions. We believe that it is better to leave the flexibility of choosing imputation methods to researchers who want to use our public dataset in the future, depending on their goals.

Based on the reasons we have stated above, we have removed the KNN imputation from the method section (page 11) and updated the corresponding figures (Figures S3-S5). We thank the reviewers for bringing this question to our attention.

4) It is essential to provide clear and detailed descriptions of the methods used in a research study, to enable reproducibility and facilitate the assessment of the validity of the results. While using a pre-existing workflow such as `bcbio` can be a useful tool for data processing and analysis, it may not provide sufficient information for readers who are not familiar with the workflow. It is therefore crucial to supplement the use of such workflows with detailed explanations of the steps taken and the parameters used, to ensure that readers can understand and reproduce the methods. Without clear and detailed descriptions, readers may not be able to assess the validity of the methods or replicate the results, which can undermine the credibility of the study. Therefore, it is important to provide a clear and thorough explanation of the methods used, regardless of the tools or workflows employed.

We thank the reviewer for this suggestion. We totally agree to provide the following details of analyzing the NGS data. All NGS data was analyzed using the best-practice workflows of the `bcbio` system. While we consider a detailed description of the workflows would exceed the space allotment of this manuscript we provide a refer to the `bcbio` documentation:

- Bulk RNA-Seq workflow: https://bcbio-nextgen.readthedocs.io/en/latest/contents/bulk_rnaseq.html
- Tumor-only variant calling workflow: https://bcbio-nextgen.readthedocs.io/en/latest/contents/somatic_variants.html

For reproducibility, we here list the versions of all tools used in this manuscript:

Software	Version
Bamtools	2.4.0
Bcbio-nextgen	1.1.0a0-7b1d1f1
Bcftools	1.7
Bedtools	2.27.1
Biobambam	2.0.87
BWA	0.7.17
CNVkit	0.9.4a0
FastQC	0.11.7
Gemini	0.20.1
Grabix	0.1.8
Kallisto	0.44.0
Qualimap	2.2.2a
Sambamba	0.6.6
Samblaster	0.1.24
Samtools	1.7
SnpEff	4.3.1t
VarDict	2018.04.27
VarDict-Java	1.5.1
VT	2015.11.10

The above information was updated in our Methods section and Supplementary Table 13.

5)The use of hypergeometric tests is a widely used approach to evaluate gene sets for enrichment analysis. While this method can be effective, it is important to consider that there are other options available for evaluating gene sets that may provide additional insights. For example, gene set enrichment analysis (GSEA) evaluates whether a set of genes shows significant differences in expression levels between two or more biological conditions, and can be useful for identifying pathways and biological processes that are differentially regulated.

We thank the reviewer for his comment. We understand the reviewer has questions on the reason why we did not use the GSEA method. It would be a good method to evaluate whether a set of genes shows significant differences in expression levels between two or more biological conditions.

However, as the reviewer also pointed out, GSEA is carried out by comparing gene expression level in two different conditions, which is before and after the cancer treatment. In our case, using the pre-treatment gene expression profile using hypergeometric test is to identify the prognostic biomarkers and provide reference for the choice of cancer treatment, which has a very different goal from GSEA.

Moreover, GSEA is usually carried out on a small number of experiments, as our high-throughput study comprises 17,912 treatment experiments, it is difficult to measure the post-treatment gene expression levels for all treatment conditions. That's why we chose hypergeometric tests to evaluate gene sets. We believe this method can be more efficient to instantly identify the pathway of interest for large-scale study, which is suitable for our situation.

However, we agree with the reviewer that GSEA can provide additional insight based on the large-scale study we carried out here. For example, the post-treatment gene expression on various cancer cell lines can be carried out for DNAPK inhibitor-gamma combination therapy and GSEA can be further carried out to identify the significantly differentially expressed genes before and after treatment. To note, the genes identified would not be prognostic biomarkers, but may unveil the more details for mode of actions for this combination treatment on cancer cell lines. We believe this could be the focus of the next steps in our research on DDR therapies.

6) What is the rationale for using the Kruskal-Wallis test, has its applicability been checked according to the conditions it should meet, and what multiple testing correction has been applied?

We thank the reviewer for his question. As we stated in the method section on page 9, Bonferroni correction was performed to adjust the multiple comparison.

Also, we checked the following conditions for Kruskal Wallist test, and made sure all conditions were met:

1. All observations from both groups are independent of each other: we confirmed this condition is met based on the design of this study. We want to compare the responses from cell lines of different cancer types. The sample is properly randomized and each experiment is carried out independently.
2. The data is ordinal: we confirmed that the data in our test, which is the responses (efficacy and synergy) in our study are continuous variables that can be meaningfully ranked.
3. The shape of distribution is the same for each group: to test if we meet this condition, I have plotted the response distributions for cancer types in the following figures.

The shape of distribution of Bliss and AoC score in different cancer types are almost identical except for bladder.

Therefore, we confirmed all conditions are met for the KW test.

7) It has been reported that Spearman's correlation coefficient was used in the analysis, which is appropriate if it has been checked that the relationship between the variables is not linear or if the data does not meet the assumptions for Pearson's correlation coefficient. Although Spearman's test is more robust against outliers, it is important to ensure that its use is justified and that the underlying assumptions are met.

We thank the reviewers for this comment. We added Pearson's correlation coefficient in the manuscript:

Interestingly, no significant correlation between average treatment efficacy or synergy and the significance of variance in different cancer types (across monotherapies (Pearson's $r = -0.01$, $p = 0.92$ and Spearman's $r = -0.075$, $p = 0.478$) and combination therapies, as well as for both efficacy (Pearson's $r = 0.01$, $p = 0.835$ and Spearman's $r = 0.01$, $p = 0.8$) and synergy (Pearson's $r = -0.04$, $p = 0.37$ and Spearman's

$r = -0.021$, $p = 0.64$) could be identified, indicating that the cancer type specificity and overall average treatment response are independent pharmaceutical characteristics. (Page 5)

8) What was the curating process used to obtain the list of 2725 genes?

We thank the reviewer for this comment. Here we provide the references of the DDR genes we obtained:

The list of investigated features covers multi-assay readouts for cancer-related genes from MSigDB¹³ with a particular focus on oncogenic signaling¹⁴ as well as genes involved in DNA damage and repair^{15–21} including also the complete Wood Lab DDR pathway collection: <https://www.mdanderson.org/documents/Labs/Wood-Laboratory/human-dna-repair-genes.html#Human%20DNA%20Repair%20Genes>. (Page 11)

It seems that sometimes there is not enough information to adequately follow the work carried out, making it difficult to fully understand the magnitude of the study. In order to properly evaluate the results and conclusions, it is important to have a clear understanding of the methods and analyses employed, as well as the limitations and potential biases that may exist. Additional details and explanations could greatly enhance the clarity and transparency of the research, and would enable readers to more effectively assess the findings and potential implications.

References :

1. Martorana, F., Da Silva, L. A., Sessa, C. & Colombo, I. Everything Comes with a Price: The Toxicity Profile of DNA-Damage Response Targeting Agents. *Cancers* **14**, (2022).
2. Mullard, A. DNA damage response drugs for cancer yield continued synthetic lethality learnings. *Nat. Rev. Drug Discov.* **21**, 403–405 (2022).
3. Mahdi, H. *et al.* Ceralasertib-mediated ATR inhibition combined with olaparib in advanced cancers harboring DNA damage response and repair alterations (Olaparib Combinations). *JCO Precis. Oncol.* **5**, 1432–1442 (2021).
4. Shah, P. D. *et al.* Combination ATR and PARP Inhibitor (CAPRI): A phase 2 study of ceralasertib plus olaparib in patients with recurrent, platinum-resistant epithelial ovarian cancer. *Gynecol. Oncol.* **163**, 246–253 (2021).
5. LaFargue, C. J., Dal Molin, G. Z., Sood, A. K. & Coleman, R. L. Exploring and comparing adverse events between PARP inhibitors. *Lancet Oncol.* **20**, e15–e28 (2019).
6. Madariaga, A., Bowering, V., Ahrari, S., Oza, A. M. & Lheureux, S. Manage wisely: poly (ADP-ribose) polymerase inhibitor (PARPi) treatment and adverse events. *Int. J. Gynecol. Cancer* **30**, 903–915 (2020).
7. Wang, C. & Li, J. Haematologic toxicities with PARP inhibitors in cancer patients: an up-to-date meta-analysis of 29 randomized controlled trials. *J. Clin. Pharm. Ther.* **46**, 571–584 (2021).
8. Coleman, R. L. *et al.* Veliparib with First-Line Chemotherapy and as Maintenance Therapy in Ovarian Cancer. *N. Engl. J. Med.* **381**, 2403–2415 (2019).
9. Yap, T. A. *et al.* Phase I Trial of First-in-Class ATR Inhibitor M6620 (VX-970) as Monotherapy or in Combination With Carboplatin in Patients With Advanced Solid Tumors. *J. Clin. Oncol.* **38**, 3195–3204 (2020).
10. Konstantinopoulos, P. A. *et al.* Berzosertib plus gemcitabine versus gemcitabine alone in platinum-resistant high-grade serous ovarian cancer: a multicentre, open-label, randomised, phase 2 trial. *Lancet Oncol.* **21**, 957–968 (2020).
11. Middleton, M. R. *et al.* Phase 1 study of the ATR inhibitor berzosertib (formerly M6620, VX-970) combined with gemcitabine±cisplatin in patients with advanced solid tumours. *Br. J. Cancer* **125**, 510–519 (2021).
12. Thomas, A. *et al.* Phase I Study of ATR Inhibitor M6620 in Combination With Topotecan in Patients With Advanced Solid Tumors. *J. Clin. Oncol.* **36**, 1594–1602 (2018).

13. Liberzon, A. *et al.* The Molecular Signatures Database (MSigDB) hallmark gene set collection. *Cell Syst* **1**, 417–425 (2015).
14. Sanchez-Vega, F. *et al.* Oncogenic Signaling Pathways in The Cancer Genome Atlas. *Cell* **173**, 321–337.e10 (2018).
15. Knijnenburg, T. A. *et al.* Genomic and Molecular Landscape of DNA Damage Repair Deficiency across The Cancer Genome Atlas. *Cell Rep.* **23**, 239–254.e6 (2018).
16. Hu, H.-M. *et al.* A Quantitative Chemotherapy Genetic Interaction Map Reveals Factors Associated with PARP Inhibitor Resistance. *Cell Rep.* **23**, 918–929 (2018).
17. Caridi, C. P. *et al.* Nuclear F-actin and myosins drive relocalization of heterochromatic breaks. *Nature* **559**, 54–60 (2018).
18. Schrank, B. R. *et al.* Nuclear ARP2/3 drives DNA break clustering for homology-directed repair. *Nature* **559**, 61–66 (2018).
19. Sundar, R., Brown, J., Ingles Russo, A. & Yap, T. A. Targeting ATR in cancer medicine. *Curr. Probl. Cancer* **41**, 302–315 (2017).
20. Paquet, N. *et al.* hSSB1 (NABP2/OBFC2B) is regulated by oxidative stress. *Sci. Rep.* **6**, 27446 (2016).
21. Bolderson, E. *et al.* Human single-stranded DNA binding protein 1 (hSSB1/NABP2) is required for the stability and repair of stalled replication forks. *Nucleic Acids Res.* **42**, 6326–6336 (2014).

REVIEWER COMMENTS

Reviewer #2 (Remarks to the Author):

The revised manuscript resolves the following concerns well:

(1) Dose-response data is now available. This is the best means of allowing future researchers to investigate key questions about efficacy at clinically relevant doses. This data will interest many in the field of combination therapy.

(2) The methods section explains that seeding densities were set to facilitate exponential growth of untreated cultures, in order to be sub-confluent by the end of the experiment. Although cell-line specific information would best support reproducibility, I agree with the authors that this is a negligible concern because, most importantly, the best technical practice has been followed.

(3) I appreciate Reviewer #1's question about toxicities, and the authors have added a balanced discussion section on the topic of combination toxicities (whose detailed study belongs to future in vivo research).

I have one remaining very major concern, which may be fixed either by editing (to remove the erroneous section), or further analysis. I respect that the authors made a good faith attempt to improve the rigor of biomarker discovery. Unfortunately the new approach to quantify False Discovery Rate is severely flawed and must not be published as-is. Subsampling is not scrambling. Subsampling 1000 times or even a billion times does not provide evidence of $FDR < 10^{-3}$ or 10^{-9} . The concept of scrambling is to randomly reassign the drug response metrics (efficacy or synergy) among cultures, so that the data is merely noise with no real signal. Then one can repeat the procedure of "discovering" biomarkers in data that is merely noise. The point of this exercise is that, because the authors have conducted a vast amount of multiple hypothesis testing (about 10,000 possible biomarker features by my understanding of the supplement), one will seem to discover strong biomarkers in data that is merely noise. If in the real data you find a particularly strong biomarker (e.g. Pearson's $r > 0.4$), you want to know how frequently would you discover an equally strong (or stronger) biomarker in the scrambled data that is merely noise. Since the articles applies rank correlation to data from 62 cell cultures this can be easily tested by simulation, which I did. For example, given the numbers of gene expression biomarkers, rank correlations of up to 0.4 are near certain by random chance alone; the same is true for LOF biomarkers, and for the geneset biomarkers, which are less numerous, correlations of up to 0.3 are near certain to be observed by random chance. Given the sample sizes in this study, it is only for rank correlations $r > 0.5$ that you reach the point where it isn't an association that is near certain to result from applying an equivalent amount of multiple hypothesis testing to random noise. The actual strengths of associations reported here are right around the magnitudes expected by simulating efforts to find Pearson correlations in random noise (given 62 cell lines and corresponding numbers of expression, LOF, and geneset biomarkers). Even correlations around >0.45 will be observed in half of random noise datasets (of corresponding size to this study). This means that the claim that biomarkers have been discovered with $FDR < 0.1\%$ (which have Pearson's r in the range of 0.3 to 0.45) is wrong by a factor of at least 100. It is because of this massive error that this section of the article should not be published.

I think the combination screen itself, augmented by the release of granular dose response data, is of sufficient interest to justify publishing a slightly trimmed-down article, without the inclusion of a flawed biomarker discovery component.

Reviewer #3 (Remarks to the Author):

I have no new concerns about this work, the authors answered properly.

RESPONSE TO REVIEWERS' COMMENTS

All reviewer's comments are marked in black
All responses are marked in blue.

Reviewer #2 (Remarks to the Author):

The revised manuscript resolves the following concerns well:

(1) Dose-response data is now available. This is the best means of allowing future researchers to investigate key questions about efficacy at clinically relevant doses. This data will interest many in the field of combination therapy.

(2) The methods section explains that seeding densities were set to facilitate exponential growth of untreated cultures, in order to be sub-confluent by the end of the experiment. Although cell-line specific information would best support reproducibility, I agree with the authors that this is a negligible concern because, most importantly, the best technical practice has been followed.

(3) I appreciate Reviewer #1's question about toxicities, and the authors have added a balanced discussion section on the topic of combination toxicities (whose detailed study belongs to future in vivo research).

Thank you.

I have one remaining very major concern, which may be fixed either by editing (to remove the erroneous section), or further analysis. I respect that the authors made a good faith attempt to improve the rigor of biomarker discovery. Unfortunately the new approach to quantify False Discovery Rate is severely flawed and must not be published as-is. Subsampling is not scrambling. Subsampling 1000 times or even a billion times does not provide evidence of $FDR < 10^{-3}$ or 10^{-9} . The concept of scrambling is to randomly reassign the drug response metrics (efficacy or synergy) among cultures, so that the data is merely noise with no real signal. Then one can repeat the procedure of "discovering" biomarkers in data that is merely noise. The point of this exercise is that, because the authors have conducted a vast amount of multiple hypothesis testing (about 10,000 possible biomarker features by my understanding of the supplement), one will seem to discover strong biomarkers in data that is merely noise. If in the real data you find a particularly strong biomarker (e.g. Pearson's $r > 0.4$), you want to know how frequently would you discover an equally strong (or stronger) biomarker in the scrambled data that is merely noise. Since the article applies rank correlation to data from 62 cell cultures this can be easily tested by simulation, which I did. For example, given the numbers of gene expression biomarkers, rank correlations of up to 0.4 are near certain by random chance alone; the same is true for LOF biomarkers, and for the geneset biomarkers, which are less numerous, correlations of up to 0.3 are near certain to be observed by random chance. Given the sample sizes in this study, it is only for rank correlations $r > 0.5$ that you reach the point where it isn't an association that is near certain to result from applying an equivalent amount of multiple hypothesis testing to random noise. The actual strengths of associations reported here are right around the magnitudes expected by simulating efforts to find Pearson correlations in random noise (given 62 cell lines and corresponding numbers of expression, LOF, and geneset biomarkers). Even correlations around >0.45 will be observed in half of random noise datasets (of corresponding size to this study). This

means that the claim that biomarkers have been discovered with $FDR < 0.1\%$ (which have Pearson's r in the range of 0.3 to 0.45) is wrong by a factor of least 100. It is because of this massive error that this section of the article should not be published.

I think the combination screen itself, augmented by the release of granular dose response data, is of sufficient interest to justify publishing a slightly trimmed-down article, without the inclusion of a flawed biomarker discovery component.

We have edited (to remove the erroneous section) as suggested.

Reviewer #3 (Remarks to the Author):

I have no new concerns about this work, the authors answered properly.

Thank you.

REVIEWERS' COMMENTS

Reviewer #2 (Remarks to the Author):

My initial review noted the interesting data on drug combinations, but noted also serious methodological concerns about the biomarker discovery sections of the study, chiefly multiple hypothesis testing on the scale of ~10,000 features with no attempt at correction or false discovery rate control, or external validation. If you were to apply the same biomarker discovery process to synthetic datasets that were entirely noise, biomarkers of similar effect sizes would be observed (for sample size matching that of this study). The revised article made a good-faith but still methodologically flawed effort to improve the biomarker discovery section, and my second review advised that the biomarker discovery section simply be removed. The authors have now removed the *revised* biomarker discovery section but the initial flawed approach and unsupported claims of biomarker discovery remain (results on pages 5 and 6, Figure 4, and many supplementary materials). This might have been a miscommunication.

At this point either of these two options would be acceptable:

(1) Remove ALL mentions of biomarker discovery from the paper. This includes Figure 4 and everything within the results subheadings

"Genomic and transcriptomic profiling identifies biomarkers for the in vitro response to DDR inhibitor combination treatment"

"Molecular markers strongly correlated with DNA-PKi-irradiation synergy"

"Molecular markers associated with DNA replication stress are strongly correlated with ATMi-ATRi synergy"

The lack of statistical rigor in all of these results sections is not acceptable for publication.

(2) Move the biomarker discovery results to the supplementary materials, and refer to them in the main text with an acknowledgement that these results stem from approximately 10,000 uncontrolled hypothesis tests, conducted in an exploratory, post-hoc manner (no particular features were pre-specified hypotheses). This is plainly true, and it will convey to readers that it may be useful data to stimulate future studies, but these correlations are not presently robust findings.

RESPONSE TO REVIEWERS' COMMENTS

All comments are marked on black.

All responses are marked in blue.

Reviewer #2 (Remarks to the Author):

My initial review noted the interesting data on drug combinations, but noted also serious methodological concerns about the biomarker discovery sections of the study, chiefly multiple hypothesis testing on the scale of ~10,000 features with no attempt at correction or false discovery rate control, or external validation. If you were to apply the same biomarker discovery process to synthetic datasets that were entirely noise, biomarkers of similar effect sizes would be observed (for sample size matching that of this study). The revised article made a good-faith but still methodologically flawed effort to improve the biomarker discovery section, and my second review advised that the biomarker discovery section simply be removed. The authors have now removed the *revised* biomarker discovery section but the initial flawed approach and unsupported claims of biomarker discovery remain (results on pages 5 and 6, Figure 4, and many supplementary materials). This might have been a miscommunication.

At this point either of these two options would be acceptable:

(1) Remove ALL mentions of biomarker discovery from the paper. This includes Figure 4 and everything within the results subheadings

"Genomic and transcriptomic profiling identifies biomarkers for the in vitro response to DDR inhibitor combination treatment"

"Molecular markers strongly correlated with DNA-PKi-irradiation synergy"

"Molecular markers associated with DNA replication stress are strongly correlated with ATMi-ATRi synergy"

The lack of statistical rigor in all of these results sections is not acceptable for publication.

(2) Move the biomarker discovery results to the supplementary materials, and refer to them in the main text with an acknowledgement that these results stem from approximately 10,000 uncontrolled hypothesis tests, conducted in an exploratory, post-hoc manner (no particular features were pre-specified hypotheses). This is plainly true, and it will convey to readers that it may be useful data to stimulate future studies, but these correlations are not presently robust findings.

We thank the reviewers for his suggestions and we apologize for the misunderstanding in the last round. We have adapted the first option by removing all mentions of biomarker discovery from the paper, including Figure4, Figure5 and results with subheadings:

1. "Genomic and transcriptomic profiling identifies biomarkers for the in vitro response to DDR inhibitor combination treatment"

2. "Molecular markers strongly correlated with DNA-PKi-irradiation synergy"

3. "Molecular markers associated with DNA replication stress are strongly correlated with ATMi-ATRi synergy"

The Materials section involves biomarker discovery, and Supplementary Tables and Figures are also removed.